# A Facile and Low-Cost Method to Produce Ultrapure 99.99999% Gallium

**DOI:** 10.3390/ma11112308

**Published:** 2018-11-17

**Authors:** Kefeng Pan, Ying Li, Jiawei Zhang, Qing Zhao

**Affiliations:** 1School of Metallurgy, Northeastern University, Shenyang 110819, China; xiaopandy@126.com (K.P.); zhang416940558@163.com (J.Z.); 18369904008@163.com (Q.Z.); 2Liaoning Key Laboratory for Metallurgical Sensors and Technology, Shenyang 110819, China

**Keywords:** gallium, metallurgy, recrystallization, impurity, segregation coefficient, purification method

## Abstract

As one of the critical raw materials, very pure gallium is important for the semiconductor and photoelectric industry. Unfortunately, refining gallium to obtain a purity that exceeds 99.99999% is very difficult. In this paper, a new, facile and efficient continuous partial recrystallization method to prepare gallium of high purity is investigated. Impurity concentrations, segregation coefficients, and the purification effect were measured. The results indicated that the contaminating elements accumulated in the liquid phase along the crystal direction. The order of the removal ratio was Cu > Mg > Pb > Cr > Zn > Fe. This corresponded to the order of the experimentally obtained segregation coefficients for each impurity: Cu < Mg < Pb < Cr < Zn < Fe. The segregation coefficient of the impurities depended strongly on the crystallization rate. All observed impurity concentrations were substantially reduced, and the purity of the gallium obtained after our refinement exceeded 99.99999%.

## 1. Introduction

Gallium (Ga), one of the important raw materials used in contemporary semiconductor industry, was discovered in 1875 [1], and has been significantly utilized in the industry since the 1940s. Ga and its compounds are extensively used in advanced electronic devices [2,3,4], integrated circuits [5,6,7], and thin-film solar cells [8,9] because these compounds can provide the benefits of low energy consumption and high computation speeds.

Current industrial production of low-grade (4N, 99.99% pure) Ga has been perfected [10,11,12]. According to a report published by the U.S. Geological Survey (USGS) in 2015, the global demand for Ga is ever-increasing and is expected to increase 20-fold by 2030 compared to the yield of 275 tons in 2012 [13]. In another statistical data reported by USGS in 2018, the world low-grade primary gallium production was estimated to be 315 tons in 2017—an increase of 15% from 274 tons in 2016. Integrated circuits accounted for 70% of domestic gallium consumption, and optoelectronic devices accounted for 30% [14]. However, even very small amounts of impurities, such as Cu, Pb, Fe, Mg, Zn, and Cr, which are always present in current large-scale, commercial-quality gallium, can degrade or limit the electrical properties [15]. Traditional refining methods, such as electrolysis and combined processes, have been used in the past to remove these impurities to obtain high-purity gallium [16]. These conventional methods, however, are very energy-consuming, harmful to the environment, and relatively slow. Hence, a superior purification method would be of great significance for the global semiconductor and photovoltaic industry.

Crystallization method is considered as the most promising technique for large-scale industrial production of high-purity Ga because of its simple equipment, ease of operation, and short production cycle. Based on the different solubility between the bulk and the contaminating elements in the liquid phase, upward directional solidification has been used to refine materials such as solar-grade silicon [17,18]. During upward directional solidification, most of the impurities with a lower segregation coefficient accumulate near the top of the ingot. Unfortunately, the low crystal growth rate usually causes visible back-diffusion in the solid phase, which reduces the purification effect [19]. In addition, low efficiency, i.e., slow growth rate, is a major problem.

In this study, continuous partial recrystallization is proposed and used to refine 4N gallium to an ultrapure level (7N, ≥99.99999% pure). The refining process was designed and optimized and the purification effect, together with contributing factors like crystallization temperature, solidification rate and ratio, were investigated. The crucial segregation coefficients of the contaminating elements in gallium were also determined.

## 2. Methods

### 2.1. Refining Process

The refining process, a schematic of which is shown in Figure 1, was implemented in a custom-made cylindrical polytetrafluoroethylene crystallizer with a jacket structure and an internal and external cavity.

Three kilograms of 4N Ga was molten at 338.15 K and washed with 150 mL, 3 mol/L HCl, and HNO_3_ for 30 min successively. In the washing process, spongy gallium was dissolved, and 2.9774 kg of liquid Ga was obtained; the percentage of gallium dissolved was 0.75%. Then, the impurity concentrations *C_0_* (wt.%) of the washed Ga were determined using glow discharge mass spectrometry (GDMS) (Evans Materials Technology (Shanghai) Co., China). Subsequently, the liquid Ga was transferred to the internal cavity. Cooling water was poured into the external cavity from the water inlet, which was located at the bottom of the crystallizer, with a 40 L/h flow rate. The water was removed through an outlet at the top. After cooling the melted Ga to 303 K, four types of crystal seeds, 0.5 cm in size, were symmetrically added along the inner wall of the inner cavity. Both temperature and flow rate of the cooling water were kept constant to allow liquid Ga to crystallize uniformly along the inner wall of the inner cavity and toward the center. When the crystallization ratio reached the preset values, the remaining liquid Ga was expelled from the discharge gate, located at the bottom center of the crystallizer, and collected as raw material for the next refining step. Subsequently, 360 K hot water was added to the external cavity to remelt the solid Ga that crystallized. The above crystallization process was repeated several times until the expected Ga purity was reached.

All refining and testing were implemented in a Class 10,000 cleanroom. Acid washing was performed using spectrally pure HCl and HNO_3_ and ultrapure water with better than 16 MΩ·cm resistivity.

### 2.2. Quality Test

The refined ultrapure Ga and remaining impurities were measured using GDMS (Evans Materials Technology (Shanghai) Co., China). A well-characterized Tantalum (Ta) check sample was used to ensure that the complete GDMS system meets the basic criteria for the required analysis. An approximately 2 cm × 0.2 cm pin of gallium was formed with liquid nitrogen. The glow discharge ion source was cooled to near liquid nitrogen temperature, and the sample was presputtered for 5 min before data acquisition began. Both data acquisition and presputtering were carried out under the same analytical conditions, and the efficiency of the ion-counting detector was checked during the analysis of the Ta Quality Control sample. Data for the gallium sample were collected until the last 3 mass fraction readings varied by no more than 20%. Elements were scanned individually, with an integration time of 80 ms. For accuracy, nine sampling points (the red dots in Figure 1) were defined to measure impurity concentration changes, while the ingot grew during the refining process.

## 3. Results and Discussion

### 3.1. Flow Rate and Temperature of the Cooling Water Influence on the Refining Process

Strict control of the solidification process during the synthesis of single crystals or polycrystalline grains with large sizes and with minimum impurities entrapped in the grain boundaries is extremely important for high-purity Ga smelting. An ideal condition is when the liquid gallium is uniformly crystallized along the inner wall of the crystallizer toward the center, and this process usually needs to be carried out in a temperature field with uniform and stable temperature gradient. For this reason, the influence of flow rate and temperature of the cooling water on the refining process was tested.

Figure 2 shows the shapes of the Ga solidification at different flow rates. It can be seen that at a flow rate of 40 L/h, the growth of Ga solidification was uniform with a direction of crystallization from the edge to the center. When the flow was changed to 60 L/h, the crystallization rate of liquid at the bottom of the crystallizer was faster than the top, and the crystal interface was rough. According to the metal crystallization theory, with the increase in cooling water flow, the overcooling degree at the bottom is greater than that in other regions and the crystallization rate is too high, resulting in excessive growth of crystals in the bottom area and uneven crystallization. In our study, at cooling water flow of 40 L/h, the growth rate on the side far away from the water inlet was far lower than that in other areas.

The crystallization rate *v* (kg/h) was controlled by changing the temperature of the cooling water between 283.15 and 297.15 K (see Figure 3). It can be seen that the crystallization rate decreased gradually with increasing temperature *T* (K). The relationship between them fit well with the following linear function:*v* = −0.09*T* + 27(1)with the linearly dependent coefficient *R*^2^ = 0.997. The strong linear correlation helps to obtain an accurate solidification ratio by controlling the crystallization time during refining.

### 3.2. Impurity Redistribution

A good understanding of the regularity of the impurity redistribution during ingot growth helps fine-tune the refining process. Unfortunately, it is difficult to obtain this important information by analyzing the morphology and microstructure via SEM [17] or other direct observation methods because of the low melting point (about 303 K) of Ga. Additionally, direct detection of the impurity concentrations in the growth direction is difficult. A convenient alternative is to measure the impurity concentrations in liquid Ga, *C_l_* (wt.%), at different solidification ratios *g* (%) during the refining process. Then, the impurity concentrations in solid Ga, *C_s_* (wt.%), can be derived from the mass conservation law using the following equation:*C_s_* = (*C*_0_ − *C_l_* · (1 − *g*))/*g*(2)

The *C_s_* profiles of the main impurities—Fe, Pb, Zn, Mg, Cu, and Cr—for a solidification ratio range of 5–95% are shown in Figure 4a. The impurity concentrations increased exponentially with increasing solidification ratios, and the trend lines were almost identical for the six elements. The results indicated that, during the refining process, the freshly solidified layer, formed at the solid/liquid interface with the Ga ingot that grows continuously from the internal wall to center of the crystallizer, transferred the impurities to the adjacent liquid layer. This way, an enriched liquid layer with impurities was created. In addition, these condensed impurities diffused towards the liquid bulk, which resulted in a progressive increase of impurities in the liquid. The difference between the impurity concentrations in solid and melt can be determined by the known equilibrium segregation coefficient, which is defined as
*K* = *C_s_*/*C_l_*(3)
for a given temperature [20]. Based on the radial exponential redistribution of each impurity in solid Ga, the effect of the solidification ratio on the removal efficiency of these impurities is defined as follows:*R*% = ((*C*_0_ − *C_s_*)/*C*_0_) · 100%(4)

When we investigated this ratio, as shown in Figure 4b, we found that *R*% for each element decreased with increase in solidification ratio, and Fe and Zn changed faster than the other four elements: Fe (86.5–25.1%), Zn (92.5–55.5%), Cr (96.9–76.2%), Pb (98.3–85%), Mg (99.2–91.1%), and Cu (99.6–94.6%). The *R*% profiles also indicated that the order of the removal radii was Cu > Mg > Pb > Cr > Zn > Fe. Based on the above results, the refining method used *R*% = 70% for the first two runs of the recrystallization process and *R*% = 85% for the third.

### 3.3. Experimental Segregation Coefficient

The effect of crystallization rate *v* on the removal ratio for the impurities was investigated for a solidification ratio of 20% (see Figure 5). The removal ratio for each element increased as the crystallization rate decreased to 0.39 kg/h. The removal ratio reached a maximum and remained stable, which indicated that the parameter of *v* also played an important role in the refining process.

The experimentally obtained segregation coefficients for each impurity were estimated as described in Reference [21] using the formula
*K_exp_* = *C_s_*/*C_l_*(5)
to study the reasons for this effect (see Figure 6a). As *v* decreased, the values for all six tested elements decreased exponentially in a crystallization rate range of 0.3–1.5 kg/h. Moreover, the three elements Fe, Pb, Zn and other group of Cr, Mg, and Cu showed a similar trend, with Fe (0.27–0.13), Pb (0.17–0.06), Zn (0.22–0.11), Cr (0.19–0.08), Mg (0.15–0.04), and Cu (0.15–0.03). Interestingly, the *K_exp_* values for different elements exhibited the same crystallization rates in the order Cu < Mg < Pb < Cr < Zn < Fe, which corresponded to the order of the removal ratios (see Figure 6b). This correlation indicated that the different removal efficiencies between different impurities were caused by the segregation coefficients in liquid Ga. Furthermore, the smaller the *K_exp_*, the better was the removal coefficient.

### 3.4. Quality of Ultrapure Gallium

Based on the above results, 3 kg of raw Ga was refined with a crystallization rate of 0.39 kg/h. The impurity concentration for each element and the purity of Ga before and after the refining process is shown in Table 1. All six tested impurity concentrations could be reduced substantially. The highest removal ratio of 93.79% was obtained for Cu, where the content decreased from 107 × 10^−6^ to 0.22 × 10^−6^ wt.%. The lowest removal ratio was 93.79% (for Fe), were the concentration decreased from 15 × 10^−6^ to 0.93 × 10^−6^ wt.%. After the refining process, 1.264 kg Ga with a purity of 99.9999958% was obtained, and all measured impurity concentration were reduced to very low levels. The remaining 1.735 kg Ga could be used as starting materials for the next refining process.

## 4. Conclusions

The optimized partial recrystallization route showed a strong purification effect: 99.99999% pure Ga was produced using little energy and a relatively simple process with the solidification ratio of 70% for the first two runs and 85% for the third run at the crystallization rate of 0.39 kg/h. During the refining process, the impurity concentrations increased exponentially with increasing solidification ratio, and the order of the removal ratio was Cu > Mg > Pb > Cr > Zn > Fe. This order corresponded to the order of the experimental values for segregation coefficient for each impurity: Cu < Mg < Pb < Cr < Zn < Fe. All tested element concentrations were reduced to very low levels, which means this novel refining method can open new doors to produce ultrapure gallium at a commercial scale.

## 5. Patent

This work was awarded a patent for invention (CN201510198211.5) by national intellectual property administration, PRC.

## Figures and Tables

**Figure 1 materials-11-02308-f001:**
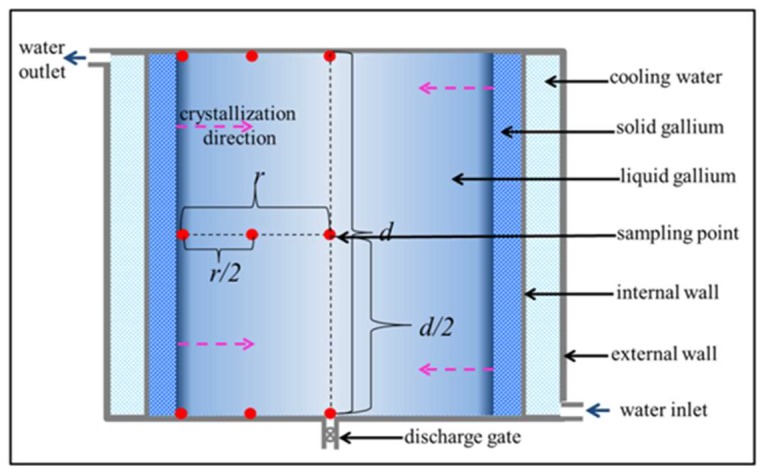
Schematic of the refining process, where *r* is the distance from the solidification interface to the center of the crystallizer, and *d* is the distance between the top and the bottom of liquid Ga.

**Figure 2 materials-11-02308-f002:**
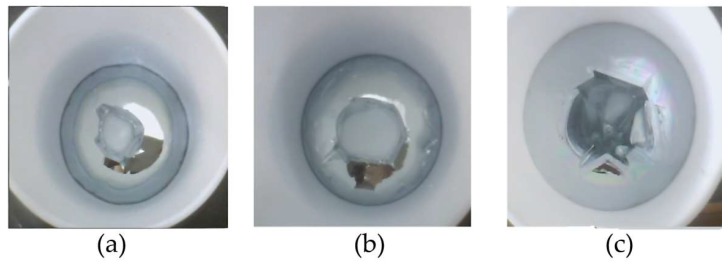
Shape of the solidification at different flow rates: (**a**) 20 L/h; (**b**) 40 L/h; and (**c**) 60 L/h in the refining process.

**Figure 3 materials-11-02308-f003:**
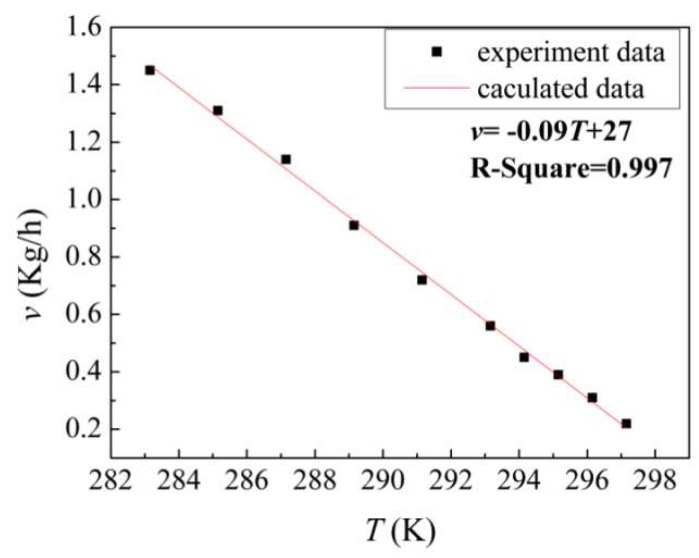
Crystallization rate *v* controlled by changing the temperature *T* of the cooling water, between 283.15 K and 297.15 K. The black points represent experimental measurements, and the red line is obtained by linear fitting.

**Figure 4 materials-11-02308-f004:**
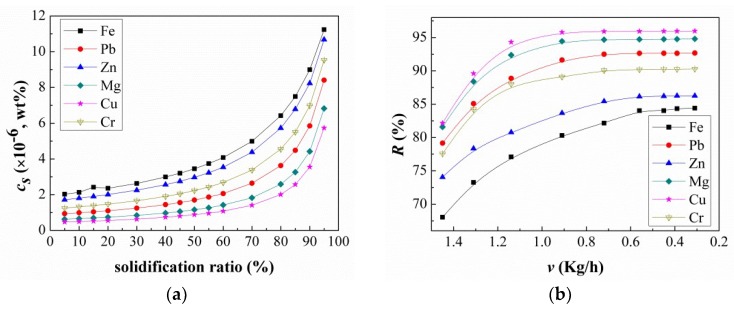
(**a**) Regularity of element impurity redistribution. *C_s_* is the impurity concentration in solid Ga. (**b**) Removal efficiency for each impurity’s *R*% at different solidification ratios.

**Figure 5 materials-11-02308-f005:**
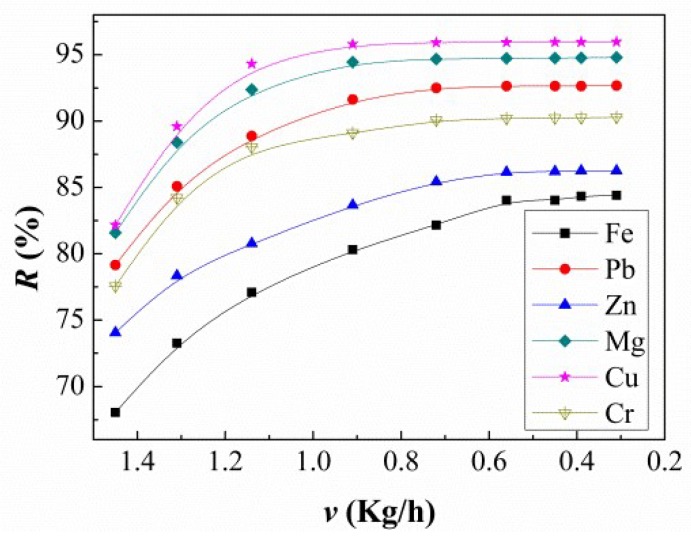
Effect of solidification rate *v* on the removal ratio for the impurities *R*%.

**Figure 6 materials-11-02308-f006:**
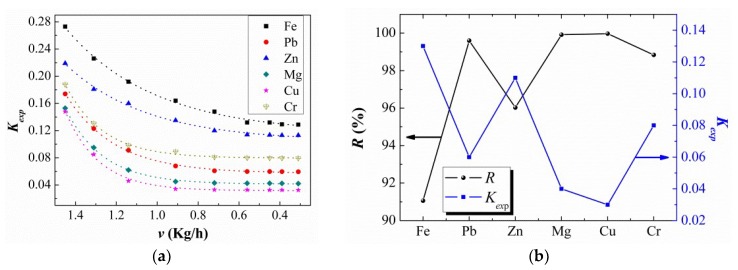
(**a**) Effect of solidification rate *v* on the experimental segregation coefficient *K_exp_*. (**b**) Relationship between the removal ratio for the impurities *R*% and the segregation coefficient *K_exp_*.

**Table 1 materials-11-02308-t001:** Impurity concentration (×10^−6^, wt.%) and purity of Ga during and after refining.

Impurity	Fe	Pb	Zn	Mg	Cu	Cr	Purity %
Before refining	15	56	24	76	107	40	99.9938
After refining	0.93	0.68	1.05	0.34	0.22	0.97	99.9999958
Removal ratio (%)	93.79	98.78	95.64	99.55	99.79	97.58	–

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
