# Peer review of "A Facile and Low-Cost Method to Produce Ultrapure 99.99999% Gallium"

_materials, 2018, doi:10.3390/ma11112308_

Reviewer 1 Report

Washing of a molted Gallium was performed using 150 mL 3mol/L HCl and HNO3.

Which percent of Gallium was dissolved during this washing? What was the washing time (1 hour?)?

Author Response

Dear Reviewer:

Thank you for your comments concerning our manuscript entitled “A Facile and Low-Cost Method to Produce Ultra-Pure 99.99999% Gallium” (ID: materials-389054). The comment is valuable and very helpful for revising and improving our paper, as well as the important guiding significance to our researches. We have studied comment carefully and have made correction which we hope meet with approval. Revised portion are marked using the “Track change” function in Microsoft Word in the paper. The main corrections in the paper and the responds to the reviewer’s comments are as following:

Comment: Washing of a molted Gallium was performed using 150 mL 3mol/L HCl and HNO3.

Which percent of Gallium was dissolved during this washing? What was the washing time (1 hour?)?

Response: We have added the descriptions about the washing time and the dissolved percent in the text: 3 kg of 4 N Ga was molten at 338.15 K and washed with 150 mL, 3 mol/L HCl and HNO3 for 30 minutes successively. In the washing process, spongy gallium was dissolved and 2.9774 kg of liquid Ga was obtained, the percent of gallium dissolved was 0.75%. Then, the impurity concentrations C0 (wt.%) of the washed Ga were determined using glow-discharge mass-spectrometry (GDMS).

We tried our best to improve the manuscript and made some changes in the manuscript. These changes will not influence the content and framework of the paper. And here we listed the changes. We appreciate for Editors/Reviewers’ warm work earnestly, and hope that the correction will meet with approval.

Once again, thank you very much for your comments and suggestions.

Reviewer 2 Report

Comments:

The authors proposed, implemented and investigated a novel method for a continuous fractional crystallization with the aim of refining 4N gallium to an ultra-pure level (7N, ≥99.99999% pure), suitable for electronic applications. In addition, the critical process parameters, e.g., crystallization temperature, solidification-rate and -ratio were investigated and the experimental segregation coefficients were determined. These contributions are of great value to research peers as well as to industrial applications.

I do, however, have a few recommendations on some specific points that could improve the text.

Line 9: add “critical” before raw materials;

Line 25: Author could replace “…significantly industrial utilized from 1940” to: …significantly utilized in industry from 1940.

Line 60: The phrase: “3 kg of 4N Ga was molten at 338.15 K and washed with 150 mL, 3 mol/L HCl and HNO3 60 successively.” gives the reader the impression that the Ga was washed and cleaned while in molten state. Is it correct? If not, I would suggest to re-write the sentence to clarify this point.

Line 82 and 88: “Ta” means Ga? If not, is it a common quality check to use a Tantalum (Ta) sample to calibrate the GDMS?

Line 114: word “fitted” was repeated twice.

Line 160 to 167: This paragraph is very difficult to understand and at the same time is delivery critical information. Therefore, I suggest improving the structure of the written text to facilitate the reading.

Line 180: Would recommend in the conclusion to mention again the final parameters and operational conditions used to achieve the results (R% = 70% for the first two runs as well as R% = 85% for the third). It would facilitate the readers to have a rapid access for this information.

The article has generally a good quality and I recommend it for publication.

Author Response

Dear Reviewer:

Thank you for your comments concerning our manuscript entitled “A Facile and Low-Cost Method to Produce Ultra-Pure 99.99999% Gallium” (ID: materials-389054). Those comments are all valuable and very helpful for revising and improving our paper, as well as the important guiding significance to our researches. We have studied comment carefully and have made correction which we hope meet with approval. Revised portion are marked using the "Track change" function in Microsoft Word in the paper. The main corrections in the paper and the responds to the reviewer’s comments are as flowing:

1. Comments: Line 9: add “critical” before raw materials.

Response: We have added “critical” in there: Very pure gallium as one of the critical raw materials is important for the semiconductor and photoelectric industry.

2. Comments: Line 25: Author could replace “…significantly industrial utilized from 1940” to: …significantly utilized in industry from 1940.

Response: We have replaced it in the text: Gallium (Ga), one of the important raw materials of contemporary semiconductor industry, was discovered in 1875 [1], and was significantly utilized in industry from 1940s.

3. Comments: Line 60: The phrase: “3 kg of 4N Ga was molten at 338.15 K and washed with 150 mL, 3 mol/L HCl and HNO3 60 successively.” gives the reader the impression that the Ga was washed and cleaned while in molten state. Is it correct? If not, I would suggest to re-write the sentence to clarify this point.

Response: Ga was exactly washed and cleaned while in molten state.

4. Comments: Line 82 and 88: “Ta” means Ga? If not, is it a common quality check to use a Tantalum (Ta) sample to calibrate the GDMS?

Response: “Ta” not means Ga, it is a common quality check to use a Tantalum (Ta) sample to calibrate the GDMS. We have added the complete spelling of Ta in the text: A well-characterized Tantalum (Ta) check sample was used to ensure that the complete GDMS system meets the basic criteria for the required analysis.

5. Comments: Line 114: word “fitted” was repeated twice.

Response: We have deleted the redundant word in the text: The relationship between them was fitted well with a linear function.

6. Comments: Line 160 to 167: This paragraph is very difficult to understand and at the same time is delivery critical information. Therefore, I suggest improving the structure of the written text to facilitate the reading.

Response: We have improving the structure of the written text: The experimentally obtained segregation coefficients for each impurity were estimated as described in Ref. [21], using the following formula:

Kexp = Cs/Cl                         (5)

to study the reasons for this effect—see Fig.6a. As v decreases, the values for all six tested elements decrease exponentially in a crystallization rate range of 0.3–1.5 kg/h. Moreover, the three elements of Fe, Pb, Zn and other group of Cr, Mg and Cu show a similar trend, respectively, Fe (0.27–0.13), Pb (0.17–0.06), Zn (0.22–0.11), Cr (0.19–0.08), Mg (0.15–0.04), Cu (0.15–0.03). Interestingly, the Kexp values for different elements exhibita same crystallization rates in the order Cu < Mg < Pb < Cr < Zn < Fe, which corresponds to the order of the removal radios—see Fig.6b. This correlation indicates that the different removal efficiencies between different impurities are caused by the segregation coefficients in liquid Ga. Furthermore, the smaller Kexp is, the better the removal coefficient is.

7. Comments: Line 180: Would recommend in the conclusion to mention again the final parameters and operational conditions used to achieve the results (R% = 70% for the first two runs as well as R% = 85% for the third). It would facilitate the readers to have a rapid access for this information.

Response: We have added the final parameters and opertaional conditions in the text: The optimized partial recrystallization route shows a strong purification effect: 99.99999% pure Ga was produced, using little energy and a relatively simple process with the solidification ratio of 70% for the first two runs as well as 85% for the third at the crystallization rate of 0.39 kg/h.

We tried our best to improve the manuscript and made some changes in the manuscript. These changes will not influence the content and framework of the paper. And here we listed the changes. We appreciate for Editors/Reviewers’ warm work earnestly, and hope that the correction will meet with approval.

Once again, thank you very much for your comments and suggestions.